# A Review of the *CACNA* Gene Family: Its Role in Neurological Disorders

**DOI:** 10.3390/diseases12050090

**Published:** 2024-05-05

**Authors:** Oliwia Szymanowicz, Artur Drużdż, Bartosz Słowikowski, Sandra Pawlak, Ewelina Potocka, Ulyana Goutor, Mateusz Konieczny, Małgorzata Ciastoń, Aleksandra Lewandowska, Paweł P. Jagodziński, Wojciech Kozubski, Jolanta Dorszewska

**Affiliations:** 1Laboratory of Neurobiology, Department of Neurology, Poznan University of Medical Sciences, 61-701 Poznan, Poland; oliszymanowicz97@gmail.com (O.S.); sanpaw2@st.amu.edu.pl (S.P.); ewepot@st.amu.edu.pl (E.P.); 86648@student.ump.edu.pl (U.G.); 86063@student.ump.edu.pl (M.K.); mg.ciaston@gmail.com (M.C.); alelew15@icloud.com (A.L.); 2Department of Neurology, Municipal Hospital in Poznan, 61-285 Poznan, Poland; adruzdz@op.pl; 3Department of Biochemistry and Molecular Biology, Poznan University of Medical Sciences, 61-701 Poznan, Poland; bslowikowski@ump.edu.pl (B.S.); pjagodzi@ump.edu.pl (P.P.J.); 4Department of Neurology, Poznan University of Medical Sciences, 61-701 Poznan, Poland; wkozubski@ump.edu.pl

**Keywords:** genetic variants, *CACNA* genes, calcium channels, neurological disease

## Abstract

Calcium channels are specialized ion channels exhibiting selective permeability to calcium ions. Calcium channels, comprising voltage-dependent and ligand-gated types, are pivotal in neuronal function, with their dysregulation is implicated in various neurological disorders. This review delves into the significance of the *CACNA* genes, including *CACNA1A*, *CACNA1B*, *CACNA1C*, *CACNA1D*, *CACNA1E*, *CACNA1G,* and *CACNA1H*, in the pathogenesis of conditions such as migraine, epilepsy, cerebellar ataxia, dystonia, and cerebellar atrophy. Specifically, variants in *CACNA1A* have been linked to familial hemiplegic migraine and epileptic seizures, underscoring its importance in neurological disease etiology. Furthermore, different genetic variants of *CACNA1B* have been associated with migraine susceptibility, further highlighting the role of *CACNA* genes in migraine pathology. The complex relationship between *CACNA* gene variants and neurological phenotypes, including focal seizures and ataxia, presents a variety of clinical manifestations of impaired calcium channel function. The aim of this article was to explore the role of *CACNA* genes in various neurological disorders, elucidating their significance in conditions such as migraine, epilepsy, and cerebellar ataxias. Further exploration of *CACNA* gene variants and their interactions with molecular factors, such as microRNAs, holds promise for advancing our understanding of genetic neurological disorders.

## 1. Introduction

Calcium ions (Ca^2+^) serve as ubiquitous second messengers in cellular signaling, playing integral roles in multiple physiological processes, including neurotransmission, muscle contraction, and gene expression regulation. Voltage-gated calcium channels (VGCCs) are crucial for regulating calcium influx into cells in response to changes in membrane potential, particularly in the nervous system. The *CACNA* gene family, which encodes the alpha subunits of VGCC complexes, is essential for forming functional calcium channels. These genes are located on different chromosomes and exhibit tissue-specific expression, particularly in the nervous system (Figure 1). VGCCs, governed by the *CACNA* gene family, play pivotal roles in neuronal excitability, synaptic transmission, and plasticity [1,2,3,4].

*CACNA1A*: The *CACNA1A* gene, located on chromosome 19p13.1, encodes the alpha-1A subunit of VGCCs. This element is predominantly expressed in neurons, particularly in the cerebellum, where it plays an essential role in synaptic transmission, neuronal excitability, and motor coordination [4].*CACNA1B*: The *CACNA1B* gene, located on chromosome 9q34.3, encodes the alpha-1B subunit of VGCCs. This subunit is primarily found in neurons and is involved in regulating calcium influx at the presynaptic terminals, modulating neurotransmitter release and synaptic transmission [3].*CACNA1C*: The *CACNA1C* gene, located on chromosome 12p13.33, encodes the alpha-1C subunit which is widely expressed in various tissues, including the brain, heart, and smooth muscle. It contributes to the formation of L-type calcium channels, which play crucial roles in cardiac and neuronal excitability, muscle contraction, and synaptic plasticity [2].*CACNA1D*: The *CACNA1D* gene encodes the alpha-1D subunit of VGCCs and is located on chromosome 3p14.3. Similar to *CACNA1C*, alpha-1D contributes to the formation of L-type calcium channels and is expressed in neurons, cardiac muscle cells, and endocrine tissues. Dysregulation of *CACNA1D* has been implicated in various neurological disorders, including epilepsy and autism spectrum disorders [5].*CACNA1E, CACNA1F, CACNA1S*: Additional members of the *CACNA* gene family include *CACNA1E, CACNA1F*, and *CACNA1S*, each responsible for encoding distinct alpha subunits expressed in specific tissues and cell types. *CACNA1E* is predominantly expressed in the brain, where it plays a role in regulating neuronal excitability. Conversely, *CACNA1F* is primarily expressed in the retina and is crucial for visual signal transduction. *CACNA1S* encodes the alpha-1S subunit found in skeletal muscle, facilitating calcium influx during excitation–contraction coupling.

The diverse members of the *CACNA* gene family contribute to calcium signaling complexity across the physiological processes, from neuronal excitability to muscle contraction and hormone secretion. Upon membrane depolarization, these channels activate, allowing Ca^2+^ influx into the cell. This triggers downstream signaling crucial for neuronal communication and synaptic transmission. VGCCs regulate neurotransmitter release at the presynaptic terminals, essential for synaptic transmission and plasticity, vital for the learning and memory mechanisms [6].

The activity of *CACNA* genes and their protein products is tightly regulated at multiple levels to ensure precise control of calcium influx and neuronal excitability. Regulation can occur through various mechanisms, including post-translational modifications, alternative splicing, and protein–protein interactions.

The transcriptional regulation of *CACNA* genes entails the coordinated actions of transcription factors and epigenetic modifications. Specific transcription factors, such as CREB (cAMP response element-binding protein) and NFAT (nuclear factor of activated T-cells), bind to DNA sequences within the promoter regions of *CACNA* genes, thereby modulating their expression in response to synaptic activity [7,8,9]. Additionally, epigenetic mechanisms, including DNA methylation and histone modifications, dynamically alter the epigenetic landscape of gene promoters, further influencing the transcription of *CACNA* genes and modifying their expression levels [10,11].

The regulation of *CACNA* genes is further complicated by post-transcriptional mechanisms. Alternative splicing of *CACNA* transcripts results in the generation of numerous isoforms with distinct functional properties, facilitating the biosynthesis of protein variants with modified voltage sensitivity or activation and inactivation kinetics [12,13]. Furthermore, microRNAs (miRNAs) play a role in post-transcriptional regulation by binding to the 3′ untranslated region (UTR) of target mRNAs, thereby modulating calcium channel expression and activity [14].

In addition to transcriptional and post-transcriptional processes, post-translational mechanisms also govern *CACNA* gene expression and functionality. The phosphorylation of *CACNA* channel subunits by protein kinases, such as protein kinase A (PKA) and protein kinase C (PKC), can alter channel activity and synaptic transmission. Furthermore, protein–protein interactions with auxiliary subunits or regulatory proteins, such as β-subunits and α2δ-subunits, impact the channel kinetics, voltage sensitivity, and plasma membrane trafficking. Moreover, activity-dependent processes, such as calcium-dependent inactivation (CDI) and calcium-dependent facilitation (CDF), dynamically modulate *CACNA* channel-gating properties in response to changes in intracellular calcium levels [15,16].

These regulatory mechanisms collectively fine-tune the expression and functionality of *CACNA* genes, ensuring the precise control of calcium influx and neuronal excitability. Dysregulation of these regulatory pathways can disturb calcium homeostasis and contribute to the pathogenesis of neurological disorders.

*CACNA* genes play pivotal roles in various neurological disorders, revealing their significance in maintaining normal brain function. In migraine, genetic changes in *CACNA1A* are associated with familial hemiplegic migraine, characterized by aura symptoms and temporary paralysis [17,18,19]. Dysfunctional calcium channels may contribute to the abnormal neuronal excitability and cortical spreading depression implicated in migraine pathogenesis. Similarly, alterations in *CACNA1* genes are linked to epilepsy types like absence seizures and juvenile myoclonic epilepsy [20,21]. Dysregulated calcium signaling can promote hyperexcitability and synchronized neuronal firing, contributing to seizure initiation and propagation. This article aims to explain how different genetic variants in the described genes may contribute to the abnormal excitability and signaling of neurons, which underlie the pathophysiology of selected diseases. Understanding the intricate roles of *CACNA* genes and their protein products in neuronal function and dysfunction is paramount for unraveling the pathophysiology of neurological disorders and devising targeted therapeutic interventions. Further exploration of the molecular mechanisms governing VGCC regulation and their implications in disease pathology holds promise for advancing our understanding and treatment of these disorders.

## 2. Clinical Consequences of *CACNA* Gene Variants

The *CACNA* gene family comprises various subunits present in the voltage-gated calcium channels, which play pivotal roles in numerous physiological processes. These channels facilitate the entry of Ca^2+^ ions into excitable cells and are implicated in various calcium-dependent functions, including muscle contraction, hormone or neurotransmitter release, and gene expression. Comprising alpha-1, beta, alpha-2/delta, and gamma subunits, calcium channels form multi-subunit complexes. The alpha-1 subunit, which forms the channel pore, governs channel activity, while the other subunits act as auxiliary components, regulating this activity [22,23]. Pathogenic variants in the *CACNA* gene can result in a range of neurological, neuropsychiatric, and neuromuscular disorders (Table 1) [24,25,26,27].

The *CACNA1A* gene encodes the calcium voltage-gated channel subunit alpha1 A, also known as subunit Cav2.1, predominantly located in the presynaptic regions of the cerebral cortex, thalamus, hypothalamus, hippocampus and cerebellum. This subunit is involved in muscle contraction, hormone or neurotransmitter release, and gene expression, by forming the conducting pore [28]. Gain-of-function mutations in Cav2.1 contribute to the development of familial hemiplegic migraine, while loss-of-function mutations lead to episodic ataxia type 2. Premature shortening of the channel protein results in seizures. Cerebellar ataxia type 6 is attributed to C-terminal polyglutamine expansion [29]. Additionally, the *CACNA1A* gene is considered a candidate gene for attention deficit hyperactivity disorder, with at least 23 pathogenic variants identified to date [28,30].

The *CACNA1B* gene encodes calcium voltage-gated channel subunit alpha1 B, also known as subunit Cav2.2, predominantly found in the presynaptic regions of the midbrain, cerebellar cells, spinal cord motor neurons, and cholecystokinin-expressing interneurons. This subunit plays a crucial role in muscle contraction, hormone or neurotransmitter release, gene expression, cell motility, cell division, cell death, and neuronal firing, by forming the conducting pore [28]. The c.4166G>A (p.Arg1389His) variant leads to a reduction in the ion current passing through the Cav2.2 channel, thereby affecting neurotransmitter release at inhibitory and excitatory synapses. Altered channel functionality results in hyperactivity resembling myoclonus and dystonia [31].

The *CACNA1C* gene encodes the calcium voltage-gated channel subunit alpha1 C, also known as subunit Cav1.2, primarily located in neuronal synapses and dendrites of the brain and cardiac muscles. This subunit plays a crucial role in maintaining synaptic plasticity, neuronal survival and fear conditioning, by forming the conducting pore [28]. Previous studies have linked single nucleotide polymorphisms (SNPs) in *CACNA1C* to diseases such as schizophrenia, depression and bipolar disorder. The rs1006737 genetic variant has been associated with altered amygdala function, mental retardation, autism, developmental disorders and cardiac arrhythmias [29]. Additionally, the *CACNA1C* gene is considered a candidate gene for attention deficit hyperactivity disorder [28].

The *CACNA1D* gene encodes calcium voltage-gated channel subunit alpha1 D, or subunit Cav1.3, primarily found in the postsynaptic regions of the brain (dendritic spines), inner hair cells, organ of Corti and heart. This subunit is involved in regulating contraction, secretion, neurotransmission and gene expression, by mediating the entry of calcium ions into excitable cells [28]. Missense mutations in Cav1.3 lead to enhanced Ca^2+^ signaling [32]. The loss of function of Cav1.3 results in congenital deafness, bradycardia and sinoatrial node (SAN) arrhythmia. Additionally, Cav1.3 plays a role in aldosterone synthesis (albeit to a lesser extent than Cav3.2), supports catecholamine release from the adrenal glands by 25%, and facilitates exocytosis in pancreatic cells. These channels play pivotal roles in shaping neuronal excitability, inducing gene transcription and promoting neuronal development. They are also involved in synaptic plasticity and memory processes [5,33].

The *CACNA1E* gene encodes the calcium voltage-gated channel subunit alpha1 E, also known as subunit Cav2.3, primarily located in presynaptic and postsynaptic regions of the hippocampus, kidney, retina, spleen and pancreatic islet cells. It plays a crucial role in neurotransmitter release and long-term potentiation by facilitating the entry of calcium ions into excitable cells [28]. At least 14 missense mutations have been identified, with 60% of the subjects exhibiting three genetic variants: c.1054G>A (p.Gly352Arg), c.2104G>A (p.Ala702Thr) and c.2101A>G (p.Ile701Val). These changes occur at the cytoplasmic ends of all S6 transmembrane segments that form the activation gate, resulting in observed gain-of-function effects. In cases where genetic changes lead to loss of channel function, a milder phenotype is experienced [34].

The *CACNA1G* gene encodes the calcium voltage-gated channel subunit alpha1 G, also known as subunit Cav3.1, primarily found in the postsynaptic regions of the cerebellum, hippocampus, thalamus, and heart. It facilitates the entry of calcium ions into excitable cells, contributing to various cellular processes including muscle contraction, hormone or neurotransmitter release, gene expression, cell motility, cell division, and cell death [28]. Adult-onset spinocerebellar ataxia (SCA42) is associated with the loss-of-function effects typically caused by positively shifting the activation threshold, particularly by the c.5144G>A (p.Arg1715His) variant [35]. In contrast, SCA42ND predominantly arises from gain-of-function mutations, notably the c.5144G>A (p.Arg1715His) missense mutation [36,37]. The variant c.6958G>T (p.Gly2320Cys) lies closest to the C-terminus of α1G, potentially explaining the mild phenotype observed (episodic vestibulocerebellar ataxia) [35].

The *CACNA1H* gene encodes the calcium voltage-gated channel subunit alpha1 H, known as subunit Cav3.2. It is located in the plasma membrane of various tissues including the brain cortex, amygdala, caudate nucleus, putamen, kidney, liver and heart. This subunit plays a role in regulating muscle contraction, secretion, neurotransmission and gene expression, by forming the pore [28]. Predominantly, the changes observed are missense mutations associated with gain-of-function, resulting in an increased influx of calcium into adrenal cells [38,39].

The *CACNA1I* gene encodes the calcium voltage-gated channel subunit alpha1 I, also known as subunit Cav3.3, found in the plasma membrane of tissues such as the cerebellum, thalamus, cerebral cortex, adrenal gland and thyroid gland. It is involved in regulating muscle contraction, hormone or neurotransmitter release, gene expression, cell motility, cell division and cell death, by forming the pore [28]. Genetic changes in this gene lead to a reduced ion current density, a right-shifted activation–deactivation voltage relationship, and slower current kinetics [40]. Additionally, these changes result in impaired sleep spindles in thalamic reticular neurons [29].

The *CACNA1S* gene encodes the calcium voltage-gated channel subunit alpha1 S, or subunit Cav1.1, present in the plasma membrane of muscles and the brain cortex. It plays a role in regulating muscle contraction by forming the pore [28]. Missense mutations in *CACNA1S* are considered risk factors for increased susceptibility to malignant hyperthermia caused by inhalational anesthetics. However, having a genetic change in this gene does not guarantee the manifestation of the disease. Approximately 1% of cases are associated with different genetic variants of *CACNA1S* [41]. Furthermore, mutations in *CACNA1S* also lead to congenital dihydropyridine receptor myopathy, characterized by dysfunction disrupting the coupling of excitation and contraction with sarcoplasmic reticulum dilatation and myofibril disorganization [42]. The pathomechanism of periodic thyrotoxic paralysis in individuals with genetic changes in *CACNA1S* remains unclear, although researchers have observed reduced ion current density and slower activation but faster deactivation of skeletal muscles in affected individuals [43].

**Table 1 diseases-12-00090-t001:** The *CACNA* gene family, related diseases and variable phenotypes.

Gene	Location of Gene	Related Diseases	Variable Phenotypes
*CACNA1A*	chromosome 19p13.13	Developmental and epileptic encephalopathy 42 (DEE42) [44]	Epileptic encephalopathy, seizures, developmental delay, intellectual disability, hyperreflexia, tremor, ataxia, athetosis, EEG abnormalities, abnormal eye movements, nystagmus [45]
		Episodic ataxia, type 2 (EA2) [44]	Ataxia, unsteadiness, vertigo, myotonia, dysarthria, migraine headache, weakness, paresthesias, interictal downbeat nystagmus, interictal vestibular dysfunction, EEG with paroxysmal activity, atrophy of cerebellar vermis, visual hallucinations, auditory hallucinations, anxiety attacks, paranoid psychosis [45]
		Migraine, familial hemiplegic (FHM1) [44]	Migraine, hemiparesis, hemiplegia, dysphasia, drowsiness, confusion, psychomotor agitation, dyscalculia, attention disturbances, impaired long-term verbal memory, cerebellar ataxia, cerebellar atrophy [45]
		Spinocerebellar ataxia 6 (SCA6) [44]	Cerebellar ataxia, dysarthria, dysphagia, cerebellar atrophy, selective loss of cerebellar Purkinje cells [45]
*CACNA1B*	chromosome 9q34.3	Neurodevelopmental disorder with seizures and nonepileptic hyperkinetic movements (NEDNEH) [46]	Developmental delay, epileptic encephalopathy, developmental regression, inability to walk, absent speech, seizures, hypsarrhythmia, hyperkinetic movements, myoclonus, dystonia, oromotor dyskinesia, choreoathetosis [47]
*CACNA1C*	chromosome 12p13.33	Neurodevelopmental disorder with hypotonia, language delay, and skeletal defects with or without seizures (NEDHLSS) [48]	Developmental delay, hypotonia, delayed walking, unsteady gait, inability to walk, speech delay, absent speech, impaired intellectual development, learning disabilities, seizures, EEG abnormalities, aggression [49]
		Timothy syndrome (TS) [48]	Developmental delay, hypotonia, impaired intellectual development, seizures, autism spectrum disorder [49]
*CACNA1D*	chromosome 3p21.1	Primary aldosteronism, seizures, and neurologic abnormalities (PASNA) [50]	Developmental delay, seizures, cerebral palsy, movement disorder [51]
*CACNA1E*	chromosome 1q25.3	Developmental and epileptic encephalopathy 69 (DEE69) [52]	Macrocephaly, poor or absent eye contact, nystagmus, roving eye movements, cortical visual impairment, arthrogryposis, congenital contractures, axial hypotonia, appendicular hypertonia, epileptic encephalopathy, globally impaired development, seizures, developmental regression after seizure onset, inability to walk, absent speech, impaired intellectual development, poor spontaneous movements, hyperkinetic movements, plastic quadriplegia, hyperreflexia, dystonia, myoclonus, EEG abnormalities, multifocal discharges, hypsarrhythmia, cortical, corpus callosum and white matter atrophy [53]
*CACNA1G*	chromosome 17q21.33	Spinocerebellar ataxia 42 (SCA42) [54]	Spinocerebellar ataxia, gait instability, dysarthria, cerebellar atrophy, nystagmus, diplopia, saccadic pursuit [54]
		Spinocerebellar ataxia 42 (early-onset, severe with neurodevelopmental disorder) (SCA42ND) [54]	Delayed psychomotor development, intellectual disability, hypotonia (axial), hyperreflexia, spasticity, poor head control, inability to walk, seizures (early-onset), syndactyly, clinodactyly, oculomotor apraxia, dysmorphic features (face) [54]
		Episodic vestibulocerebellar ataxia type 10 (EVCA10) [35]	Episodes of transient dizziness, gait unsteadiness, a sensation of fall triggered by head movements, headache, cheek numbness, visual blurring, mental slowing and fatigue, reduced vestibulo-ocular reflex (VOR) [34]
*CACNA1H*	chromosome 16p13.3	Epilepsy (childhood absence, susceptibility to 6) [55]	Febrile seizures, myoclonic/astatic epilepsy, idiopathic generalized epilepsy, generalized epilepsy, childhood absence epilepsy, febrile seizures, temporal lobe epilepsy [56]
		Epilepsy (idiopathic generalized, susceptibility to 6) [57]	
		Hyperaldosteronism (familial, type IV) (HALD4) [55]	Hypertension, microscopic hyperplasia of adrenal gland glomerulosa, elevated aldosterone levels, low renin levels [58]
*CACNA1I*	chromosome 22q13.1	Neurodevelopmental disorder with speech impairment and with or without seizures (NEDSIS) [59]	Short stature, cortical visual impairment, feeding difficulties, gastroesophageal reflux, tube feeding, global developmental delay (profound), hypotonia, inability to walk, delayed walking, impaired intellectual development (profound), absent speech, seizures, myoclonus, staring spells, hyperexcitability, EEG abnormalities, cortical atrophy, delayed myelination, hypogenesis of the corpus callosum [60]
*CACNA1S*	chromosome 1q32.1	Malignant hyperthermia susceptibility to, 5 (MHS5) [61]	Hyperthermia [62]
		Thyrotoxic periodic paralysis, susceptibility to, 1 (TTPP1) [61]	Tremor, hypo- or areflexia during attacks, muscle paralysis and weakness (episodic), muscle aches, cramps [62]
		Congenital myopathy 18 (CMYP18) [61]	Delayed motor development, hypotonia, generalized muscle weakness, muscle atrophy [62]

## 3. *CACNA* Genes and Migraine

Migraine is a prevalent neurological disorder that significantly impacts public health [63]. A meta-analysis by Woldeamanuel and Cowan in 2017 revealed migraine’s prevalence across European countries to be as high as 11.4% [64]. Notably, migraine is a chronic condition that often affects individuals during their peak years of professional activity, disproportionately impacting women with a prevalence three times higher than in men [65]. The solid and uncontrolled attacks of headaches experienced by patients lead to a significantly diminished quality of life, as they are excluded from family, professional, and social life [66]. Moreover, migraine poses a formidable challenge to the medical community due to its complex pathomechanism, the necessity for timely and accurate diagnosis, and the development of effective treatment strategies.

According to the International Classification of Headache Disorders, 3rd edition (ICHD-3), migraine manifests as unilateral, pulsating headaches of moderate to severe intensity lasting from 4 to 72 h. These headaches are often accompanied by symptoms such as nausea, sensitivity to sound (phonophobia), and sensitivity to light (photophobia). Furthermore, migraine attacks may be preceded by transient neurological symptoms, primarily visual disturbances, known as visual aura, lasting usually less than 60 min. Clinically, migraine can be categorized into two main types: migraine without aura (MO) and migraine with aura (MA) [67,68].

Migraine is recognized as a multifactorial, polygenic disorder. However, the precise pathomechanism underlying migraine pain remains incompletely understood. Numerous hypotheses have been proposed to elucidate the pathogenesis of migraine pain, including the cortical spreading depression (CSD) theory. According to this theory, migraine may occur due to ion channel dysfunction, implicating channelopathy in its etiology [69,70].

The cortical spreading depression (CSD) hypothesis posits that migraine pain arises from a slow wave of depolarization in neurons and glial cells, followed by their prolonged suppression [71]. It is believed that cortical spreading depression occurs due to an increased release of the neurotransmitter glutamate into the synaptic cleft, the increased stimulation of N-methyl-D-aspartate receptors (NMDAr) and an elevated concentration of Ca^2+^ ions [72]. This process suggests that the initiation and propagation of CSD requires the influx of Ca^2+^ ions through presynaptic P/Q-type Ca^2+^ channels. These channels play a crucial role in regulating the level of Ca^2+^ ions and are encoded by genes belonging to the *CACNA* family (Figure 2) [73].

### 3.1. CACNA1A Gene in Migraine

The *CACNA1A* gene, responsible for encoding the alpha-1 subunit of the CaV2.1 calcium channel, holds significant importance in discussions surrounding migraine genetics [74]. Genetic changes within this gene have been identified as direct contributors to familial hemiplegic migraine type I (FHM1), a rare subtype of migraine characterized by its autosomal dominant inheritance pattern. FHM1 typically presents with paresis during migraine attacks, referred to as movement aura. While transient hemiparesis constitutes the classic motor aura in FHM1, a range of other neurological symptoms may be present including dysphasia, sensory loss, visual disturbances, and dizziness. These deficits often persist longer than the accompanying headache [75,76,77].

While the precise pathophysiological events in FHM1 remain incompletely understood, studies suggest that different genetic variants of the *CACNA1A* gene may alter the excitability of the neuronal CaV2.1 channels and increase the probability of channel opening, resulting in persistent gain-of-channel effects [78]. Single-channel studies have revealed that CaV2.1 channels with FHM1 mutations tend to activate at more negative membrane potentials, thereby increasing calcium influx from the extracellular space into neurons in response to action potentials in active zones of nerve endings. Consequently, CaV2.1 channels exhibit a state of sustained facilitation, remaining open for longer durations [79]. Moreover, the heightened activity of CaV2.1 channels leads to an increased influx of calcium into the presynaptic terminals, thereby promoting elevated glutamate release, a critical neurotransmitter, into the extracellular compartment [80]. This results in enhanced synaptic transmission at the excitatory synapses compared to the inhibitory synapses [78]. The increase in excitatory synaptic activity leads to cortical spreading depression (CSD) followed by a wave of synchronized depolarization that is responsible for the aura observed during FHM migraine attacks [78,81]. Thus, it becomes evident that the hyperexcitability observed in FHM1 directly stems from the augmented synaptic activity mediated by the CaV2.1 channels, the subunits encoded by the *CACNA1A* gene.

With the pivotal role of the *CACNA1A* gene in the pathogenesis of migraine headaches, particularly in familial hemiplegic migraine type I (FHM1), modern research is increasingly directed towards examining the effects of mutations within this gene on calcium channel function, along with the identification of novel mutations. Among the more than 25 different genetic variants identified in the *CACNA1A* gene associated with FHM1, the most common are the missense variants such as T666M, R192Q, and S218L [77,82]. The T666M variant involves the substitution of threonine with methionine at position 666 in the protein sequence of the L-type calcium channel subunit. Similarly, the R192Q variant involves replacing the amino acid arginine with glutamine at position 192 in the protein sequence, while the S218L variant leads to the substitution of serine at position 218 with leucine [83,84,85]. 

### 3.2. Other CACNA Genes in Migraine

The discovery of novel genetic variants of the *CACNA1A* gene associated with familial hemiplegic migraine type I (FHM1) has sparked further investigation into the variants of genes within the *CACNA* family among individuals suffering from headaches.

Given the considerable attention garnered by the *CACNA1A* gene in FHM1, emerging research suggests the potential involvement of other genes within the CACNA family in the manifestation of migraine pain. The expression patterns of the *CACNA* family genes are notably diverse, with all members except *CACNA1S* expressed in the brain, suggesting their potential involvement in headache development [86]. For instance, research by Rasmussen et al. [87] has highlighted *CACNA1B*, encoding Cav2.2, as a gene frequently mutated in families afflicted by migraine, with molecular changes observed in 16 out of 117 studied families. Additionally, Ambrosini et al. [88] identified a polymorphism in exon 20 of the *CACNA1E* gene (rs35737760), which was significantly more prevalent in patients experiencing complex neurological aura compared to controls. Furthermore, a study by Kurtuncu et al. [89] from 2013 suggests a potential immunological influence on headache development involving genes from the *CACNA* family. Kurtuncu et al. [89] identified antibodies against *CACNA1H* in two patients with HaNDL (a syndrome of transient headache and neurological deficits with cerebrospinal fluid lymphocytosis), considered a rare variant of migratory headache.

In summary, exploring the role of *CACNA* family genes represents a novel direction for further research on the molecular mechanisms of migraine. The discovery of various variants of *CACNA* genes, including *CACNA1B* and *CACNA1E*, suggests that the entire *CACNA* gene family may play a crucial role in the pathogenesis of migraine headaches. Further research is needed to fully understand the mechanisms in which *CACNA* family genes are involved.

## 4. *CACNA* Gene Variants and the Development of Epilepsy

Epilepsy, as defined by the International League Against Epilepsy (ILAE), is a neurological disorder characterized by a heightened susceptibility to recurrent epileptic seizures and their associated consequences. The diagnosis of epilepsy necessitates the presence of at least one epileptic seizure, which arises from abnormal synchronized neuronal activity, resulting in transient neurological symptoms [90]. The age-standardized incidence of epilepsy was 621.5 per 100,000 population; 540.1–737.0 [91]. Moreover, the highest incidence of epilepsy was observed among the elderly [92]. Notably, status epilepticus, a state of prolonged seizure activity, is most frequently reported in infants and the elderly [93].

The etiology of seizures may vary based on age demographics. In children, seizures are predominantly attributed to genetic factors, perinatal injuries and developmental abnormalities of the cerebral cortex [94]. Conversely, among adults, structural or genetic causes are more common. Traumatic brain injuries, brain tumors, brain malformations, strokes, vascular defects and perinatal injuries are the primary structural abnormalities associated with epilepsy in adults [95]. In elderly individuals, cerebrovascular diseases, neurodegenerative diseases, brain tumors and traumatic injuries are the predominant causes of epilepsy [96]. Notably, gender does not appear to exert a significant influence on the incidence of epilepsy. Despite extensive research, the etiopathology of epilepsy remains unknown in over 50% of cases (Figure 2) [92].

### 4.1. CACNA1A Gene in Epilepsy

The correlation between the *CACNA1A* genetic variants and the occurrence of epilepsy seems to have been well documented for quite a long time (Table 2) [97,98,99]. Furthermore, there is evidence suggesting that epileptic encephalopathy may be the result of a de novo *CACNA1A* mutation [100]. In cases of epilepsy in children associated with the *CACNA1A* variant, focal seizures predominantly occurred, possibly followed by unilateral brain atrophy. The early onset of seizures and developmental delay was noted [101]. Mutations in the *CACNA1A* gene can lead to an early impairment in P/Q release, manifesting the epileptic phenotype through the destabilization of thalamocortical rhythms [102]. Notably, in a mouse model, the loss of P/Q solely in Purkinje cells has been observed to induce absence epilepsy [103].

In studies examining epileptogenesis, a reduction in *CACNA1A* levels was observed through Western blot and histological analyses. Additionally, it was noted that curcumin administration alleviated these changes in *CACNA1A* protein expression [104]. Furthermore, there is evidence indicating that the homozygous variant c.6975_6976insCAG may underlie progressive myoclonic epilepsy [105]. Interestingly, reports have surfaced regarding asymptomatic carriers of monoallelic *CACNA1A* variants [106]. Mutations can lead to either gain of function (GOF) or loss of function (LOF). GOF mutations are characterized by intractable, early and recurrent status epilepticus, while LOF mutations are associated with refractory and early absence seizures [107]. Status epilepticus linked to GOF mutations is more frequently observed in the p.Val1392Met variant [98]

### 4.2. Other CACNA Genes in Epilepsy

It has been demonstrated that bi-allelic loss-of-function variants encoded by *CACNA1B, CACNA2D1* and *CACNA2D2* in various components of N-type calcium channels are associated with epileptic and developmental encephalopathy (Table 2) [108]. Mice with a mutation in the *CACNA1G* gene experienced spike-wave seizures due to the overexpression of ⍺1G and increased T-type calcium currents [109]. In a cohort study involving Japanese and Hispanic patients, a potential association of *CACNA1G* with idiopathic epilepsy subsyndromes was identified [110]. Variants in *CACNA1H* may be linked to temporal lobe epilepsy, febrile seizures, childhood absence, juvenile absence, juvenile myoclonus and myoclonic astatic epilepsies [111]. *CACNA1H* polymorphisms may contribute to absence epilepsy through CaV3.2 channel hyperactivity [112]. However, it appears unlikely that the *CACNA1H* mutation is solely responsible for epilepsy in humans [113].

**Table 2 diseases-12-00090-t002:** The role of the *CACNA* gene in the pathogenesis of epilepsy.

Gene	Genetic Variants	Protein Product	A Type of Epileptic Seizure	Cohort	References
*CACNA1A*	c.2963_2964insG c.3089 + 1G>A, c.4755 + 1G>T, c.6340-1G>A	p.Gly989Argfs*78	Absence epilepsy Partial epilepsy Epileptic encephalopathy	318 cases with partial epilepsy 150 cases with generalized epilepsy 296 healthy volunteers	[20]
	c.203G>T c.3965G>A c.5032C>T c.5393C>T	p.Arg68Leu p.Gly1322Glu p.Arg1678Cys p.Ser1798Leu			
	c.4891A>G c.5978C>T c.3233C>T c.6061G>A	p.Ile1631Val p.Pro1993Leu p.Ser1078Leu p.Glu2021Lys			
*CACNA1A*	c.T677G	p.Leu226Trp	Juvenile myoclonic epilepsy	Family cases	[21]
*CACNA1A*	c.301G>C c.653C>T c.2137G>A c.2137G>A c.4531G>T	p.Glu101Gln p.Ser218Leu p.Ala713Thr p.Ala713Thr	Epilepsy of infancy with migrating focal seizures Early-onset epileptic encephalopathy	531 individuals	[100]
*CACNA1A*	c.2137G>A c.4177G>A c.2039-2040del c.3968G>A c.2131G>A c.185A>G c.4177G>A c.2276T>C c.4406C>T c.4177G>A c.165A>C c.2053C>T c.6530-1G>C c.4177G>A c.889G>A c.4177G>A c.506G>A c.848A>G	p.A713T p.V1393M p.Gln680ArgfsTer100 p.G1323E p.A711T p.Y62C p.V1393M p.I759T p.S1469L p.V1393M p.R55S p.Q685X ---- p.V1393M p.G297R p.V1393M p.W169X p.N283S	Focal, Generalized tonic Clonic, Myoclonic Absence seizures Epileptic spasms Tonic seizures	Eighteen children	[101]
*CACNA1A*	c.6975_6976insCAG	insertion at amino acid position 2326	Progressive myoclonic epilepsy	Family cases	[105]
*CACNA1G*	c.1709C>T c.3265G>T c.2968G>A	p.Ala570Val p.Ala1089Ser p. Asp980Asn	Juvenile myoclonic epilepsy	123 patients with idiopathic generalized epilepsies	[110]
*CACNA1H*	c.2626G>A c.2947G>A c.3175G>T c.3508G>A c.3792G>T c.4817C>T c.5113G>A c.5197A>G c.5675G>A	p.Ala876Thr p.Gly983Ser p.Ala1059Ser p.Glu1170Lys p.Gln1264His p.Thr1606Met p.Ala1705Thr p.Thr1733Ala p.Arg1892His	Childhood absence epilepsy Febrile seizures Temporal lobe epilepsy Myoclonic-astatic epilepsy Symptomatic generalized epilepsy Juvenile myoclonic epilepsy Idiopathic generalized epilepsy Juvenile absence epilepsy	240 epilepsy patients 95 control subjects	

## 5. *CACNA* Genes and Cerebellar Ataxias

Cerebellar ataxias (CAs) may have numerous causes, including genetic factors. Autosomal recessive CAs are most frequently caused by mutations in genes such as genes associated with Friedreich ataxia (*FXN*) and associated with ataxia-telangiectasia (*ATM*). Mutations in genes such as *ATXN1, ATXN2, ATXN3, CACNA1A, ATXN7, TBP* or *ATN1* are the primary causes of autosomal dominant CAs [114]. *CACNA1A* mutations are known to underlie two CAs: episodic ataxia type 2 (EA2) and spinocerebellar ataxia type 6 (SCA6) [115]. Additionally, mutations in *CACNA1G* have been associated with a rare form of CA known as spinocerebellar ataxia type 42 (SCA42), as well as a case of episodic vestibulocerebellar ataxia reported in a single family (Figure 2).

### 5.1. Episodic Ataxia Type 2 (EA2)

Episodic ataxias (EAs) encompass a group of rare autosomal dominant hereditary diseases, with an estimated incidence of less than 1 per 100,000 individuals, although this figure may be underestimated [116]. Among EAs, EA2 stands out as the most prevalent subtype [115,117]. The earliest documentation of EA2 dates back to 1946, with its genetic association to *CACNA1A* published in 1996 [118]. EA2 typically arises from loss-of-function variants, often involving nonsense or frameshift mutations in the *CACNA1A* gene. These mutations disrupt the function of the pore-forming α1A subunit of the CaV2.1 (P/Q-type) Ca^2+^ channel encoded by *CACNA1A* [115,116,117]. CaV2.1 channel expression is mostly restricted to neuroendocrine cells and neurons throughout the nervous system [119], with abundant expression in cerebellar Purkinje cells and granule layer neurons [117].

The hallmark presentation of EA2 includes intermittent episodes of ataxia and dysarthria lasting several hours, sometimes extending up to 2–3 days. Interictal nystagmus, a characteristic feature of EA2, might help in distinguishing it from other EAs [116]. Accompanying symptoms during attacks can encompass: diplopia, tinnitus, dystonia, nausea, emesis, hemiplegia, and headache, including migraines [115,116]. Given that *CACNA1A* mutations are also implicated in FHM1 and SCA6, there is often overlap among these disorders. Despite differing genetic mechanisms (FHM1 typically involves missense gain-of-function mutations), migraine is present in up to half of the patients suffering from EA2 [115,116,117]. Triggers for attacks include: stress, caffeine, alcohol, exertion, fever, heat, and phenytoin. Attack frequency varies widely, from one or two per year to even several attacks per day. The onset of classic spells typically occurs in childhood or adolescence, although uncharacteristic symptoms such as delayed walking and speaking, cognitive impairment and autism may appear at a very early age, preceding typical attacks [115,116].

The first-line treatment in EA2 is acetazolamide, a carbonic anhydrase inhibitor, which can reduce or completely abolish attacks. Second-line treatments include 4-aminopyridine and flunarizine [115,116]. Although over 80 *CACNA1A* gene variants linked to EA2 have been identified, no genotype–phenotype relationships have been established. In 1996, the first two variants associated with EA2 were identified in two unrelated patients, both resulting in frameshift mutations. The first mutation involves a single C deletion at nucleotide 4073 in codon 1266, leading to a frameshift in the putative translation product with a premature stop codon in the next exon. The second mutation is a G-to-A transition at the first nucleotide of intron 24, disrupting the highly conserved GT dinucleotide at the intronic 5-prime splice junction and resulting in the loss of a BsaAI restriction site [118].

### 5.2. Spinocerebellar Ataxia Type 6 (SCA6)

Spinocerebellar ataxias (SCAs) represent a group of progressive and degenerative genetic disorders characterized by an unsteady gait along with poorly coordinated speech, hand, and eye movements [120]. The overall prevalence of SCA is estimated to be around 50 per 100,000 individuals, with SCA6 being one of the most prevalent subtypes [121]. The genetic association of SCA6 with the *CACNA1A* gene was first published in 1997 [122]. The onset of SCA6 typically occurs later than that of EA2, usually between the ages of 43 and 52, although onset can range from 19 to 71 years [121]. Initial symptoms often include gait unsteadiness, stumbling and balance issues, with the disease progressing gradually. As the condition advances, patients may develop symptoms such as gait ataxia, upper limb incoordination, intention tremors, dysarthria, visual impairments (including diplopia and nystagmus) or difficulties swallowing [120,121]. In patients with SCA6, in vivo MRI scans have revealed not only damage to the cerebellum but also degeneration in the cerebral cortex, thalamus, midbrain, pons, and medulla [120]. At present, there are no effective treatments for SCA6.

The underlying cause of SCA6 is an expanded CAG(n) repeat within exon 47 of the *CACNA1A* gene, which encodes a polyglutamine tract. Normal alleles typically contain 4 to 18 repeats, whereas pathogenic alleles contain 19 to 33 repeats. The length of the expansion correlates with the age of disease onset, with longer expansions associated with earlier onset [120]. Unlike EA2 or FHM1, the pathophysiology of SCA6 does not directly affect channel kinetics. Within *CACNA1A*, there exists an internal ribosome entry site that initiates the translation of a second peptide, a1ACT, containing the expanded polyglutamine tract implicated in SCA6. The a1ACT peptide functions as a transcription factor, regulating the expression of key genes involved in synaptic formation, neurogenesis, and cell adhesion [115]

### 5.3. Spinocerebellar Ataxia Type 42 (SCA42)

Spinocerebellar ataxia type 42 (SCA42) manifests as an autosomal dominant adult-onset neurodegenerative disorder characterized by gradually progressive ataxia alongside other cerebellar symptoms. This disorder was initially described by Morino et al. [123], who investigated two Japanese families exhibiting dominant traits for cerebellar ataxia. Family 1 comprised ten affected individuals, while Family 2 had five affected individuals. The clinical presentation resembled that of SCA6, with the onset of symptoms ranging from 18 to 70 years. Alternative causes of the symptoms were ruled out, leading to the identification of mutations in the *CACNA1G* gene. Specifically, the mutation p.Arg1715His was located at S4 of repeat IV, the voltage sensor of the CaV3.1 channel. CaV3.1 channels are expressed in the cerebellum, inferior olive nucleus, and thalamus [123].

### 5.4. Episodic Vestibulocerebellar Ataxia

Furthermore, Gazulla et al. [124] established a connection between episodic vestibulocerebellar ataxia and a missense variant in *CACNA1G*. Within a family, two individuals experienced episodes consisting of vertigo, headache, unsteady gait, and other symptoms. The heterozygous *CACNA1G* missense variant c.6958G>T (p.Gly2320Cys) was discovered in symptomatic individuals and was not discovered in a healthy family member who was also tested. The authors proposed naming this disease episodic ataxia type 10 [124].

## 6. *CACNA* Genes and Cerebellar Atrophy and Other Neurological Diseases

Cerebellar atrophy refers to the chronic and irreversible decline in neuronal structure and function within the cerebellum, with Purkinje cells being particularly susceptible. The causes of cerebellar atrophy can generally be categorized as acquired or genetic. Acquired cerebellar atrophy is attributed to various endogenous or exogenous non-genetic factors, including vitamin deficiencies, alcohol abuse, infections of the central nervous system, autoimmune disorders and primary or metastatic tumors, among others [125]. Diseases that cause cerebellar atrophy my also involve other regions of the nervous system such as the spinal cord, cerebral cortex and brainstem. Neurological diseases that feature cerebellar atrophy include stroke, Friedreich ataxia and other spinocerebellar ataxias, transmissible spongiform encephalopathies (for example Creutzfeldt–Jakob disease) and multiple sclerosis. Progressive disorders characterized by cerebellar atrophy as a prominent feature include cerebellar cortical atrophy, multisystem atrophy, and olivopontocerebellar degeneration [126].

Symptoms of cerebellar atrophy may include limb ataxia, gait and stance ataxia, dysarthria, oculomotor disturbance and non-motor deficits affecting executive functions, working memory, language, visuospatial cognition and social behavior. Studies show that cerebellar atrophy leads to specific impairments in various motor skill learning tasks, such as visuomotor adaptation and adaptation to external forces. Moreover, implicit motor sequence learning and visuomotor associative learning are impaired [127].

Cerebellar atrophy imposes a significant health burden on the global population. Cerebellum-related disorders typically manifest between the ages of 45 to 65, although the age of symptomatic onset varies across different diseases. For instance, paraneoplastic cerebellar degeneration is more prevalent in women and typically begins around the age of 50. In cases of alcoholic cerebellar atrophy, symptoms usually manifest in middle-aged individuals with a history of chronic alcohol abuse, often due to thiamine deficiency, which is also implicated in nutritional cerebellar atrophy (Figure 2) [128].

### Genetic Variants of CACNA1A

Pathogenic variants in the *CACNA1A* gene have been linked to various neurological disorders, including familial hemiplegic migraine and cerebellar conditions. Within the central nervous system, *CACNA1A* is highly expressed and encodes the pore-forming CaVα1 subunit of P/Q-type (Cav2.1) calcium channels [129]. FHM represents an autosomal dominant subtype of migraine with aura. While pure FHM exhibits genetic heterogeneity, cases featuring cerebellar manifestations often stem from mutated *CACNA1A*. Examples of such a patient include a proband from France. The individual experienced recurrent, severe, and prolonged episodes of hemiplegic migraine, alongside mental retardation and early-onset disabling ataxia with cerebellar atrophy. Genetic screening uncovered a new de novo missense mutation in the *CACNA1A* gene. Despite a normal birth, she did not achieve the ability to sit until 2.5 years of age and could not walk until reaching 7 years old. Upon assessment, she displayed severe cerebellar gait ataxia, bilateral limb incoordination, slight dysarthria, and gaze-evoked nystagmus. Her first migraine attack occurred at age 7, followed by subsequent episodes at ages 8, 25, 31 and 33. During the third attack, she experienced rapid onset mental obtundation, vomiting, photophobia, right hemiplegia, and fever. A brain MRI performed after a week revealed diffuse left cortical abnormalities suggestive of edema, right cerebral hemisphere atrophy, and cerebellar atrophy. Fifteen days later, a subsequent contrast angiography showed multiple segmental narrowing of intracranial branches of the left carotid and basilar artery, along with vasodilation of the distal branches of the left hemisphere arteries. Consciousness and temperature resolved after 2 days, with motor deficits returning to pre-attack levels after four weeks. Her parents had never experienced a migraine attack and their neurologic examination was without any abnormalities. DNA of the proband and her parents was extracted from peripheral blood, with additional access to the DNA of 100 unrelated healthy individuals. A comprehensive screening of all 47 exons of the *CACNA1A* gene was conducted using a combination of single-strand conformer polymorphism and sequencing analysis. To confirm the de novo nature of the mutation and eliminate any possibility of false paternity, the proband and her parents underwent genotyping with three intragenic markers specific to CACNA1A, along with eight polymorphic CA repeats located on different chromosomes. Among the *CACNA1A* intragenic markers were the GCG repeat in exon 1, D19S1150 (a CA repeat within intron 7), and the CAG repeat in exon 47. Additionally, eight other markers were utilized to establish paternity and maternity: D1S233, D3S1192, D5S407, D6S282, D7S479, D9S152, D10S537, and D14S80. The analysis revealed a novel missense mutation in exon 26, where an A/G substitution at codon 1385 (TAC/TGC) resulted in the replacement of a tyrosine with a cysteine. In a panel of 200 normal chromosomes, this genetic change did not emerge. Additionally, a sequence analysis of exon 26 in both parents showed homozygosity for the normal sequence (TAC/TAC at codon 1385). Thus, this study facilitated the rejection of false paternity and led to the conclusion that this genetic change occurred spontaneously (de novo). Several compelling arguments support the notion that the Tyr1385Cys variant is responsible for the hemiplegic migraine syndrome observed in this patient, indicating its deleterious nature and ruling out the possibility of it being a polymorphism. Firstly, this change was not found in the 200 healthy chromosomes screened. Secondly, neither of the parents carried the changed variant. Thirdly, the genetic change affects segment 5 of the third domain of the channel, which is a crucial functional domain highly conserved and involved in calcium selectivity. This domain has been shown to be affected in previously reported families with FHM. Additionally, the severe cerebellar atrophy observed in the patient is likely linked to the Tyr1385Cys variant, considering that 20% of FHM families suffer from permanent cerebellar signs and the length of the CAG repeat fell within the normal range, ruling out SCA6. Moreover, P/Q-type voltage-dependent calcium channels are the primary calcium channels in Purkinje cells [130].

Different genetic variants of the *CACNA1A* gene are not only implicated in causing more severe cerebellar atrophy, but can also be associated with early onset. For instance, a 3-year-old boy with early cerebellar atrophy displayed moderate global developmental delay, persistent ataxia and nystagmus. His prenatal and perinatal period proceeded without any notable events. Concerns arose for his parents when he failed to independently sit up at 7 months of age. Thereafter his milestones were relatively delayed. He also exhibited a mild delay in speech development, primarily affecting his expressive language. By the age of 3, he frequently experienced falls and displayed instability in his movements. At 29 years old, his mother was in good health, without any personal or familial medical history. His father, aged 37, experienced occasional headaches in his late teens and early twenties, but they did not warrant medical consultation. His headaches did not follow the classic pattern of migraine attacks: there was no preceding aura or neurological deficit, they lasted around 30 min to an hour, and were easily relieved by simple painkillers or rest. These headaches disappeared entirely around the age of 30. Additionally, he had no history of developmental delays during childhood and was otherwise in good health. Further exploration of the family history revealed that 11 distant relatives on the paternal side experienced severe recurrent headaches, and one of them was clinically diagnosed with familial hemiplegic migraine at the age of 7 years. The boy’s MRI revealed moderate, generalized cerebellar atrophy with no detectable parenchymal signal changes, showing slightly more pronounced effects in the cerebellar vermis. The brainstem and cerebral hemispheres appeared normal. The patient has been regularly reviewed by neurologists and has shown good developmental progress. His cerebellar signs had stabilized by the age of 4.5 years. Although repeat cranial imaging was considered, it was deemed unnecessary due to his good progress. Given the combination of cerebellar atrophy and a family history of migraine or hemiplegic migraine, investigation into a possible mutation in the *CACNA1A* gene was explored. Sequence analysis confirmed heterozygosity for the variant c.1997C>T (p.T666M). Subsequently, the father was also tested and found to be heterozygous for the same mutation. The distant relatives, who exhibited the typical clinical presentation of familial hemiplegic migraine, were also examined and verified to possess the same heterozygous variant. This genetic change, widely recognized in cases of familial hemiplegic migraine, results in the substitution of threonine with methionine, unequivocally indicating its pathogenic nature. This case demonstrates the diversity of phenotypes linked to mutations in the *CACNA1A* gene. The patient’s delayed motor milestones and early onset of nystagmus imply that cerebellar atrophy likely existed at birth, suggesting that, in certain instances, congenital cerebellar atrophy may be linked to a mutation in this gene. Therefore, mutations in the *CACNA1A* gene should be taken into account in the differential diagnosis of cerebellar atrophy in pediatric patients, irrespective of their history of familial hemiplegic migraine or episodic ataxia. [131].

Genetic alterations in the *CACNA1A* gene, owing to its high expression in the central nervous system, can lead to various neurological conditions. Rarely, it may manifest as a combination of cerebellar ataxia and isolated cervical dystonia, as evidenced by a 62-year-old man with a history of gait instability. Despite a normal physical examination, neurological assessment revealed characteristic symptoms, including ataxic gait and cerebellar atrophy detected via MRI. Genetic testing identified a heterozygous variant in the *CACNA1A* gene, resulting in cervical dystonia and cerebellar ataxia [132]. The detected variant was a one-nucleotide insertion c.4056_4057insG (p.Pro1353Alafs*3), leading to a reading frame shift resulting in a premature stop codon three amino acids after the insertion. Studies in mice with *CACNA1A* mutations further support this association, emphasizing the gene’s role in cerebellar development and neuron survival [133].

## 7. Conclusions

Ion channels, including VGCCs, are fundamental to various physiological processes in the nervous system. Among the diverse subtypes of VGCCs, the *CACNA* gene family is one of the most essential. Ongoing research utilizing modern molecular techniques has facilitated the identification of numerous *CACNA* types, ranging from *CACNA1A-1I* and *CACNA1S*, each with varying implications in the development of neurological diseases such as migraine, epilepsy, ataxia, dystonia, and cerebellar degeneration and different clinical trajectories.

Furthermore, it is important to highlight that the activity of *CACNA* genes and their protein products is tightly regulated at multiple levels to ensure proper neuronal function. This regulation can occur through a variety of mechanisms, including post-translational modifications, alternative splicing, and protein–protein interactions.

Understanding novel *CACNA* variants and the molecular-level changes associated with their occurrence holds the potential to provide fresh insights into the pathogenesis of diseases where calcium channels play a pivotal role.

## Figures and Tables

**Figure 1 diseases-12-00090-f001:**
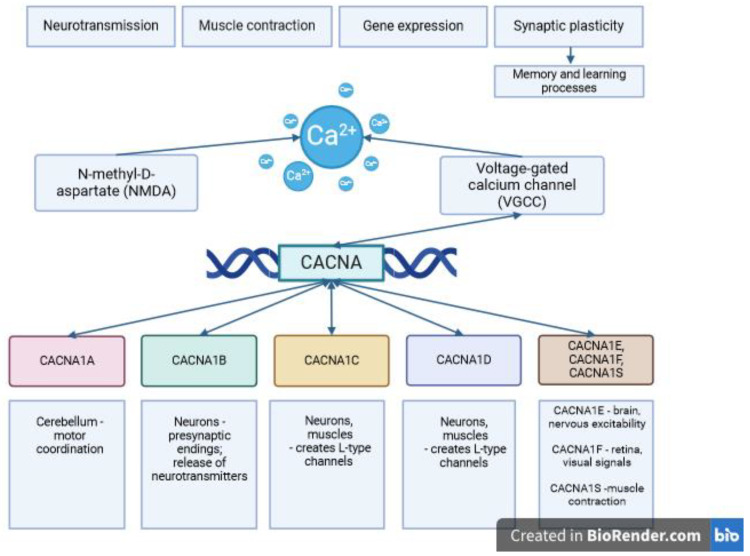
The role of *CACNA* genes in the functioning of the nervous system. The function of individual *CACNA* from 1A to 1F, and 1S genes in the nervous system and the linkage of *CACNA* genes to the mechanisms that regulate Ca^2+^ ion transport.

**Figure 2 diseases-12-00090-f002:**
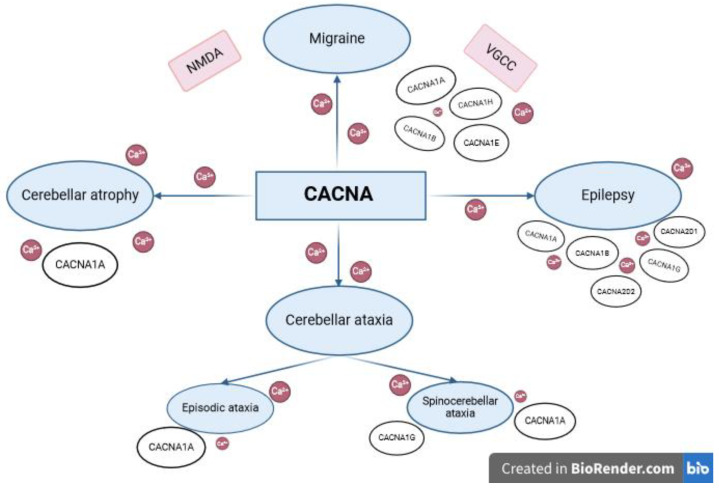
Involvement of *CACNA* genes in the pathogenesis of neurological diseases. The involvement of variants of the Ca^2+^ channel gene, *CACNA*, in the etiology of rare neurological diseases, the common denominators of which are variable paroxysmal symptoms (migraine, epilepsy) and chronic cerebellar dysfunction (cerebral atrophy, cerebral ataxia).

## Data Availability

Not applicable.

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
