# Peer review of "A Review of the CACNA Gene Family: Its Role in Neurological Disorders"

_diseases, 2024, doi:10.3390/diseases12050090_

Round 1
Reviewer 1 Report
Comments and Suggestions for Authors
Dear Editor
The manuscript "The role of the CACNA gene in neurology" is very well written. However, there are several of the concerns that needs to be addressed.
1. Title of the study is not much informative as this is difficult to understand from title either it is "research article or review".
2. Abstract section does not contain the need of the project, rather it has long discussion on the background.
3. Introduction section: no clear hypothesis or need of the article is added.
4. The figures does not show sufficient data.
5. The tables are presented well.
Overall, the authors are requested to add the critical need and future perspectives of this article.
Thanks and Regards
Author Response
Thank you for the detailed and substantive improvement of the manuscript.
In reply to the comments of Reviewer I;
Comments and Suggestions for Authors
The manuscript "The role of the CACNA gene in neurology" is very well written. However, there are several of the concerns that needs to be addressed.
- Title of the study is not much informative as this is difficult to understand from title either it is "research article or review".
The title of the article has been changed
,,A Review of CACNA Gene Family: Their Role in Neurological Disorders"
- Abstract section does not contain the need of the project, rather it has long discussion on the background.
The abstract has been corrected, the purpose of the article and the future perspective have been emphasized:
Page 1, lines 14-29
Calcium channels are specialized ion channels exhibiting selective permeability to calcium ions. Calcium channels, comprising voltage-dependent and ligand-gated types, are pivotal in neuronal function, with their dysregulation implicated in various neurological disorders. This review delves into the significance of CACNA genes, including CACNA1A, CACNA1B, CACNA1C, CACNA1D, CACNA1E, CACNA1G, and CACNA1H, in the pathogenesis of conditions such as migraine, epilepsy, cerebellar ataxia, dystonia, and cerebellar atrophy. Specifically, variants in CACNA1A have been linked to familial hemiplegic migraine and epileptic seizures, underscoring its importance in neurological disease etiology. Furthermore, mutations in CACNA1B have been associated with migraine susceptibility, further highlighting the role of CACNA genes in migraine pathology. The complex relationship between CACNA gene variants and neurological phenotypes, including focal seizures and ataxia, presents a variety of clinical manifestations of impaired calcium channel function. The aim of this article was to explore the role of CACNA genes in various neurological disorders, elucidating their significance in conditions such as migraine, epilepsy, and cerebellar ataxias. Further exploration of CACNA gene variants and their interactions with molecular factors, such as microRNAs, holds promise for advancing our understanding of genetic neurological disorders.
- Introduction section: no clear hypothesis or need of the article is added.
The purpose of this article has been added:
Page 3, lines 119-126
This article aims to explain how mutations, in the described genes, may contribute to abnormal excitability and signaling of neurons, which underlies the pathophysiology of selected diseases. Understanding the intricate roles of CACNA genes and their protein products in neuronal function and dysfunction is paramount for unraveling the pathophysiology of neurological disorders and devising targeted therapeutic interventions. Further exploration of the molecular mechanisms governing VGCC regulation and their implications in disease pathology holds promise for advancing our under-standing and treatment of these disorders.
- The figures does not show sufficient data.
This article contains two figures, created by the authors, describing the role of CACNA genes in the functioning of the nervous system (Figure 1.) and involvement of CACNA genes in the pathogenesis of neurological diseases (Figure 2.).
The first figure titled ,,The role of CACNA genes in the functioning of the nervous system” shows the function of individual CACNA genes in the nervous system and the linkage of CACNA genes on the mechanisms that regulate calcium ion transport. The second figure titled ,,Involvement of CACNA genes in the pathogenesis of neurological diseases” presents the importance of the individual CACNA genes described in selected neurological diseases.
It seems that the prepared figures are sufficient for this article.
Added
Page 4, line 130-132
Figure 1. The role of CACNA genes in the functioning of the nervous system. The function of individual CACNA from 1A to 1F, and 1S genes in the nervous system and the linkage of CACNA genes to the mechanisms that regulate calcium ion transport.
Page 19, lines 610-612
Figure 2. Involvement of CACNA genes in the pathogenesis of neurological diseases. The involvement of variants of the Ca2+ channel gene, CACNA, in the etiology of rare neurological diseases, the common denominator of which are variable paroxysmal symptoms (migraine, epilepsy) and chronic cerebellar dysfunction (cerebral atrophy, cerebral ataxia).

Reviewer 2 Report
Comments and Suggestions for Authors
Try to adjust the titles in the paper.
Use recent references.
You can make the introduction shorter if it possible.
You can modify the type of the table if you want.
Use pictures and illustrations in the paper
Author Response
Thank you for the detailed and substantive improvement of the manuscript.
- Try to adjust the titles in the paper.
The title of the article has been changed:
,,A Review of CACNA Gene Family: Their Role in Neurological Disorders"
- Use recent references.
References have been updated:
- Amiri, P.; Kazeminasab, S.; Nejadghaderi, S.A.; Mohammadinasab, R.; Pourfathi, H.; Araj-Khodaei, M.; Sullman, M.J.M.; Kolahi, A.A., Safiri, S. Migraine: A Review on Its History, Global Epidemiology, Risk Factors, and Comorbidities. Front. Neurol. 2022, 12, 800605.
- Rossi, M.F.; Tumminello, A.; Marconi, M.; Gualano, M.R.; Santoro, P.E.; Malorni, W.; Moscato, U. Sex and gender differences in migraines: a narrative review. Neurol. Sci. 2022, 9, 5729-5734.
- Weyrer, C.; Turecek, J.; Niday, Z.; Liu, P.W.; Nanou, E.; Catterall, W.A.; Bean, B.P.; Regehr, W.G. The Role of CaV2.1 Channel Facilitation in Synaptic Facilitation. Cell Rep. 2019, 26, 2289-2297.
- Cárdenas-Rodríguez. N.; Carmona-Aparicio, L.; Pérez-Lozano, D.L.; Ortega-Cuellar, D.; Gómez-Manzo, S.; Ignacio-Mejía, I. Genetic variations associated with pharmacoresistant epilepsy (Review). Mol. Med. Rep. 2020, 21, 1685-1701.
- Gandini, M.A.; Souza, I.A.; Ferron, L.; Innes, A.M.; Zamponi, G.W. The de novo CACNA1A pathogenic variant Y1384C associated with hemiplegic migraine, early onset cerebellar atrophy and developmental delay leads to a loss of Cav2.1 channel function. Mol. Brain. 2021, 14, 27.
- You can make the introduction shorter if it possible.
The introduction has been revised and condensed for clarity.
- You can modify the type of the table if you want.
In the authors' opinion, the tables prepared are transparent and clear, and the data contained in them are clearly visible.
- Use pictures and illustrations in the paper
This article contains two figures, created by the authors, describing the role of CACNA genes in the functioning of the nervous system and involvement of CACNA genes in the pathogenesis of neurological diseases. In addition, there are two tables describing the CACNA gene family, related diseases and variable phenotypes and the role of the CACNA gene in the pathogenesis of epilepsy. It seems that the number of figures and tables is sufficient for this article.
Added descriptions:
Page 4, line 130-132
Figure 1. The role of CACNA genes in the functioning of the nervous system. The function of individual CACNA from 1A to 1F, and 1S genes in the nervous system and the linkage of CACNA genes to the mechanisms that regulate calcium ion transport.
Page 19, lines 610-612
Figure 2. Involvement of CACNA genes in the pathogenesis of neurological diseases. The involvement of variants of the Ca2+ channel gene, CACNA, in the etiology of rare neurological diseases, the common denominator of which are variable paroxysmal symptoms (migraine, epilepsy) and chronic cerebellar dysfunction (cerebral atrophy, cerebral ataxia).

Reviewer 3 Report
Comments and Suggestions for Authors
I enclose my comments

Author Response
Thank you for the detailed and substantive improvement of the manuscript.
In reply to the comments of Reviewer III;
- Introduction
- Long introduction
The introduction has been revised and condensed for clarity.
- Comment regarding the line 49 and the Table 1.
The sentence on line 49 has been corrected.
Page 1 lines 40-43
These genes are located on different chromosomes and exhibit tissue-specific expression, particularly in the nervous system (Fig. 1). VGCCs, governed by the CACNA gene family, play pivotal roles in neuronal excitability, synaptic trans-mission, and plasticity [1-4].
Page 7, line from 237
Genes localization has been added to the Table 1.
|
Gene |
Location of gene |
Related diseases |
Variables phenotypes |
|
CACNA1A
|
chromosome 19p13.13 |
Developmental and epileptic encephalopathy 42 (DEE42) [44]
Episodic ataxia, type 2 (EA2) [44]
Migraine, familial hemiplegic (FHM1) [44]
Spinocerebellar ataxia 6 (SCA6) [44] |
Epileptic encephalopathy, seizures, developmental delay, intellectual disability, hyperreflexia, tremor, ataxia, athetosis, EEG abnormalities, abnormal eye movements, nystagmus [45]
Ataxia, unsteadiness, vertigo, myotonia, dysarthria, migraine headache, weakness, paresthesias, interictal downbeat nystagmus, interictal vestibular dysfunction, EEG with paroxysmal activity, atrophy of cerebellar vermis, visual hallucinations, auditory hallucinations, anxiety attacks, paranoid psychosis [45]
Migraine, hemiparesis, hemiplegia, dysphasia, drowsiness, confusion, psychomotor agitation, dyscalculia, attention disturbances, impaired long-term verbal memory, cerebellar ataxia, cerebellar atrophy [45]
Cerebellar ataxia, dysarthria, dysphagia, cerebellar atrophy, selective loss of cerebellar Purkinje cells [45] |
|
CACNA1B
|
chromosome 9q34.3 |
Neurodevelopmental disorder with seizures and nonepileptic hyperkinetic movements (NEDNEH) [46] |
Developmental delay, epileptic encephalopathy, developmental regression, inability to walk, absent speech, seizures, hypsarrhythmia, hyperkinetic movements, myoclonus, dystonia, oromotor dyskinesia, choreoathetosis [47] |
|
CACNA1C
|
chromosome 12p13.33 |
Neurodevelopmental disorder with hypotonia, language delay, and skeletal defects with or without seizures (NEDHLSS) [48]
Timothy syndrome (TS) [48]
|
Developmental delay, hypotonia, delayed walking, unsteady gait, inability to walk, speech delay, absent speech, impaired intellectual development, learning disabilities, seizures, EEG abnormalities, aggression [49]
Developmental delay, hypotonia, impaired intellectual development, seizures, autism spectrum disorder [49] |
|
CACNA1D
|
chromosome 3p21.1 |
Primary aldosteronism, seizures, and neurologic abnormalities (PASNA) [50] |
Developmental delay, seizures, cerebral palsy, movemnt disorder [51] |
|
CACNA1E
|
chromosome 1q25.3 |
Developmental and epileptic encephalopathy 69 (DEE69) [52] |
Macrocephaly, poor or absent eye contact, nystagmus, roving eye movements, cortical visual impairment, arthrogryposis, congenital contractures, axial hypotonia, appendicular hypertonia, epileptic encephalopathy, globally impaired development, seizures, developmental regression after seizure onset, inability to walk, absent speech, impaired intellectual development, poor spontaneous movements, hyperkinetic movements, pastic quadriplegia, hyperreflexia, dystonia, myoclonus, EEG abnormalities, multifocal discharges, hypsarrhythmia, cortical, corpus callosum and white matter atrophy [53] |
|
CACNA1G |
chromosome 17q21.33 |
Spinocerebellar ataxia 42 (SCA42) [54]
Spinocerebellar ataxia 42 (early-onset, severe with neurodevelopmental disorder) (SCA42ND) [54]
Episodic vestibulocerebellar ataxia type 10 (EVCA10) [35]
|
Spinocerebellar ataxia, gait instability, dysarthria, cerebellar atrophy, nystagmus, diplopia, saccadic pursuit [54]
Delayed psychomotor development, intellectual disability, hypotonia (axial), hyperreflexia, spasticity, poor head control, inability to walk, seizures (early-onset), syndactyly, clinodactyly, oculomotor apraxia, dysmorphic features (face) [54]
Episodes of transient dizziness, gait unsteadiness, a sensation of fall triggered by head movements, headache, cheek numbness, visual blurring, mental slowing and fatigue, reduced vestibulo-ocular reflex (VOR) [34] |
|
CACNA1H
|
chromosome 16p13.3 |
Epilepsy (childhood absence, susceptibility to 6) [55] Epilepsy (idiopathic generalized, susceptibility to 6) [56]
Hyperaldosteronism (familial, type IV) (HALD4) [55]
|
Febrile seizures, myoclonic/astatic epilepsy, idiopathic generalized epilepsy, generalized epilepsy, childhood absence epilepsy, febrile seizures, temporal lobe epilepsy [57]
Hypertension, microscopic hyperplasia of adrenal gland glomerulosa, elevated aldosterone levels, low renin levels [58] |
|
CACNA1I
|
chromosome 22q13.1 |
Neurodevelopmental disorder with speech impairment and with or without seizures (NEDSIS) [59]
|
Short stature, cortical visual impairment, feeding difficulties, gastroesophageal reflux, tube feeding, global developmental delay (profound), hypotonia, inability to walk, delayed walking, impaired intellectual development (profound), absent speech, seizures, myoclonus, staring spells, hyperexcitability, EEG abnormalities, cortical atrophy, delayed myelination, hypogenesis of the corpus callosum [60] |
|
CACNA1S
|
chromosome 1q32.1 |
Malignant hyperthermia susceptibility to, 5 (MHS5) [61]
Thyrotoxic periodic paralysis, susceptibility to, 1 (TTPP1) [61]
Congenital myopathy 18 (CMYP18) [61] |
Hyperthermia [62]
Tremor, hypo- or areflexia during attacks, muscle paralysis and weakness (episodic), muscle aches, cramps [62] Delayed motor development, hypotonia, generalized muscle weakness, muscle atrophy [62] |
- Comment to the 52-134 lines
In the indicated lines there is a short description of selected CACNA genes, their location and function in the neurological system, which refers to the later Figure 1. The whole is only a beginning of the description of the importance of individual CACNA genes in selected neurological diseases, therefore the authors believe that this fragment is part of introduction.
- Clinical consequences of CACNA gene variants
- The acronymus NMDA in line 273 and Figure 1. has been described. The legend of the Figure 1. has been corrected.
The cortical spreading depression (CSD) hypothesis posits that migraine pain arises from a slow wave of depolarization in neurons and glial cells, followed by their prolonged suppression [71]. It is believed that cortical spreading depression occurs due to increased release of the neurotransmitter glutamate into the synaptic cleft, in-creased stimulation of N-methyl-D-aspartate receptors (NMDAr) and an elevated concentration of Ca2+ ions [72]. This process suggests that the initiation and propagation of CSD requires the influx of Ca2+ ions through presynaptic P/Q-type Ca2+ channels. These channels play a crucial role in regulating the level of Ca2+ ions and are encoded by genes belonging to the CACNA family (Fig. 2) [73].
Figure 1. The role of CACNA genes in the functioning of the nervous system. The function of individual CACNA from 1A to 1F, and 1S genes in the nervous system and the linkage of CACNA genes to the mechanisms that regulate Ca2+ ion transport.
- Comment to the line 148
The sentence has been corrected:
Pathogenic variants in the CACNA gene can lead to a spectrum of neurological, neuropsychiatric, and neuromuscular disorders (Table 1) [24-27].
In the manuscript, each phrase variation has been replaced by variants.
In the article the word mutation has been replaced with other descriptions. In some cases, the word mutation was retained to reflect the nature of the change.
- Comment to the line 174
Yes, the literature we know proves evidence for the establishment of CACNA1C (rs1006737 variant) as a susceptibility gene for schizophrenia across world populations.
Reference in the article:
- Zhu, D.; Yin, J.; Liang, C.; Luo, X.; Lv, D.; Dai, Z.; Xiong, S.; Fu, J.; Li, Y.; Lin, J.; Lin, Z.; Wang, Y.; Ma, G. CACNA1C (rs1006737) may be a susceptibility gene for schizophrenia: An updated meta-analysis. Brain Behav. 2019, 9, 01292.
- Comments to the Table 1.
Table 1. is extensive due to the large amount of data that had to be presented briefly. The font has been reduced.
The title of the Table 1. has been changed. The headings in the table columns have been changed.
Table 1. The CACNA gene family, related diseases and variable phenotypes
|
Gene |
Location of gene |
Related diseases |
Variables phenotypes |
- Section from 3 to 6
- Comment to the Table 2.
Table 2. summarizes the information contained in the paragraph 4. CACNA gene variants and the development of epilepsy. The Table 2. makes numerous genes and their genetic variants clear and easy to use.
- Comments to the section from 2 to 6
- Starting from section 2. the importance of the CACNA1A gene has been described in migraine, epilepsy and ataxia. Additionally, paragraph 6.1. describes various genetic variants of the CACNA1A
- The introduction has been limited.
- Sections 6.1. and 6.2. have been combined and shortened. Now section 6.1. describes genetic variants of CACNA1A
- Various CACNA genes are listed and described both in the introduction and in the remaining sections describing neurological diseases.
- Within the literature examined, much attention has been given to the CACNA1A gene, given its established role in familial hemiplegic migraine (FHM). However, recent progress in genetic research has led to the discovery of new CACNA genes that play important roles in the pathophysiology of the selected diseases. We would like to emphasize the importance of genes from the CACNA family, responsible for calcium metabolism, in the pathophysiology of neurological diseases.

Round 2
Reviewer 1 Report
Comments and Suggestions for Authors
Dear Editor
The manuscript is significantly updated with incorporation of suggested revisions.
Comments on the Quality of English LanguageDear Editor
Minor errors could be corrected during MDPI proof-read.
Reviewer 3 Report
Comments and Suggestions for Authors
-